# Carrageenan-Based Films Incorporated with Jaboticaba Peel Extract: An Innovative Material for Active Food Packaging

**DOI:** 10.3390/molecules25235563

**Published:** 2020-11-27

**Authors:** Luisa Bataglin Avila, Elis Regina Correa Barreto, Paloma Krolow de Souza, Bárbara De Zorzi Silva, Thamiris Renata Martiny, Caroline Costa Moraes, Marcilio Machado Morais, Vijaya Raghavan, Gabriela Silveira da Rosa

**Affiliations:** 1Engineering Graduate Program, Federal University of Pampa, 1650 Maria Anunciação Gomes de Godoy Avenue, Bagé, Rio Grande do Sul 96413-172, Brazil; luisabataglinavila@gmail.com (L.B.A.); thamiris.martiny@hotmail.com (T.R.M.); 2Chemical Engineering, Federal University of Pampa, 1650 Maria Anunciação Gomes Godoy Avenue, Bagé, Rio Grande do Sul 96413-172, Brazil; elisrbarretoeq@gmail.com (E.R.C.B.); palomakrolow@gmail.com (P.K.d.S.); barbara.zorzi@hotmail.com (B.D.Z.S.); marciliomorais@unipampa.edu.br (M.M.M.); 3Chemical Engineering Department, Federal University of Santa Maria, Santa Maria, Rio Grande do Sul 97105-900, Brazil; 4Graduate Program in Science and Engineering of Materials, Federal University of Pampa, 1650 Maria Anunciação Gomes de Godoy Avenue, Bagé, Rio Grande do Sul 96413-172, Brazil; caroline.moraes@unipampa.edu.br; 5Department of Bioresource Engineering, McGill University, 21111 Lakeshore Road, Ste-Anne-de-Bellevue, QC H9X 3V9, Canada; vijaya.raghavan@mcgill.ca

**Keywords:** bioactive compounds, antioxidant, microwave-assisted extraction, carrageenan film, jaboticaba peel

## Abstract

This research investigated the bioactive potential of jaboticaba peel extract (JPE) and proposed an innovative material for food packaging based on carrageenan films incorporated with JPE. The extract was obtained through microwave assisted extraction (MAE) according to central composite rotational design and the optimized conditions showed a combined antimicrobial and antioxidant actions when the extraction process is accomplished at 80 °C and 1 min. The carrageenan film incorporated with JPE was manageable, homogeneous and the presence of JPE into film increased the thickness and improved the light barrier of the film. The results of solubility and mechanical properties did not show significant differences. The benefit of using MAE to improve the recovery of bioactive compounds was demonstrated and the carrageenan film with JPE showed a great strategy to add additives into food packaging.

## 1. Introduction

Jaboticaba is a highly perishable fruit native to Brazil and belongs to the *Myrtacea* family, botanically classified as *Myrciaria* spp. This fruit is widely consumed in Brazil and is popularly known as Brazilian Grape, since its popularity is compared to that of grapes in the United States. The great acceptance of jaboticaba in the consumer market can be due to its pleasant taste and its properties, such as the presence of vitamins, flavonoids, and anthocyanins, as well as its antioxidant effects against free radicals [1]. Despite the many functional properties of the jaboticaba fruit, during industrial processes, like the production of juices, jams, and liquors, a lot of jaboticaba by-products are discarded, such as its peels, seeds, and a portion of the pulp. The by-products are not added to the final product, even though they contain most of the fruit’s nutrients and bioactive compounds. Therefore, better use of these residues might add more value to the raw material [2].

Recent research efforts focused on the recovery of these valuable compounds and even more in their use as additives in food, nutraceuticals, and pharmaceuticals. The literature reports that the study of extraction conditions is an important factor for the recovery of bioactive compounds from plant materials. Parameters such as solvent type, temperature, and extraction time, as well as extraction method (conventional or clean techniques) are cited, as critical factors for the extraction of these compounds [3,4].

Therefore, the use of alternative methods is of great importance to overcome these disadvantages. As previously reported, microwave-assisted extraction (MAE) presents efficient extraction yields in a shorter time, without degradation of the extracted bioactive compounds [5]. MAE utilizes the microwave effect by heating up the internal moisture of plant cells, resulting in its evaporation, and generating pressure on cell walls. The generated pressure leads to the eruption of cell walls, leading to an increase in the yield of phytoconstituents through exudation of the bioactive constituents from the ruptured cells walls [4].

These bioactive compounds extracted from the jaboticaba by-products could be incorporated into biodegradable films to maximize their additional characteristics, such as their antimicrobial and antioxidant capacities. Different types of natural polymers are used to produce biodegradable films as an alternative to the use of synthetic polymers; among these are proteins and polysaccharides such as carrageenans [6,7]. Carrageenans are polysaccharides that are considered to be natural water-soluble polymers. This biopolymer is composed of a linear chain of sulfated galactans and is extracted from seaweeds of the Rhodophyceae family [8,9,10]. Additionally, carrageenans are attractive for their high potential as a film-forming material [6,9,11].

Many studies were published about the development of biodegradable films incorporating natural compounds that use several polymeric matrices, including carrageenan. Adilah et al. [12] revealed the benefits of mango by-products incorporated into gelatin-based films, as a potential material for active packaging. Chi et al. [13] produced a κ-carrageenan-based intelligent film incorporating grape skin powder, and Martiny et al. [14] developed an active biodegradable film incorporated with olive leaves. Despite all these advances, the literature did not show results of studies using carrageenan biodegradable films containing jaboticaba peel extract, in order to determine alternatives to chemical food preservatives and eco-friendly packaging, aiming to increase the shelf life of food products.

The incorporation of a natural bioactive compound for the production of a packaging based on a biodegradable polymer matrix represents a direction for both the substitution of non-biodegradable plastics and the extension of the shelf life of food products, without compromising the quality of the same through addition of chemical preservatives. The development of carrageenan biodegradable films incorporated with jaboticaba peels extracts would be an alternative to active packaging for foods, since the antioxidant and antimicrobial actions of the extract would inhibit oxidative processes and the growth of microorganisms, reducing the deterioration of food, and thus increasing its shelf life. In addition, the film could be used as a substitute for conventional plastic packaging—an environmental problem. In the present context, the development of films from carrageenans and jaboticaba peels extract, presented in this innovative and unprecedented research, could present a great potential in the diversification of the formulation of films.

The aim of this innovative research was to evaluate the use of an environment-friendly extraction, using microwave-assisted extraction (MAE); to optimize the process parameters for the recovery of bioactive compounds from jaboticaba peels; and to develop carrageenan-based biodegradable films by incorporating jaboticaba peel extract (JPE).

## 2. Results and Discussion

### 2.1. Extracts Optimization and Characterization

The extracts produced showed different shades, according to the experimental condition. The same difference in shade was reported by Gurak et al. [2], in which depending on the technological property employed, the colors of the jaboticaba extracts varied.

The total phenolic compounds (TP), antioxidant activity (AA), total anthocyanin (TA), and cyanidin 3-glucoside (C-3-G) are shown in Table 1. The results confirmed that JPE has an active potential, with a high content of phenolic compounds and antioxidant activity. In comparison, Quatrin et al. [15] reported 246.5 mg_GAE_·g^−1^ of total phenolic compounds, using a methanol/water/formic acid solution and sonication for exhaustive extraction of the jaboticaba peels. Faria et al. [16], found a 97% of antioxidant activity in the freeze-dried peel using a methanol solution, and found a 95% antioxidant activity, using hydroalcoholic solution. Calloni et al. [17] reported 242.75 mg·g^−1^ for cyanidin 3-glucoside content of the peel extracted with distilled water, under reflux. Lima et al. [18] evaluated the extraction of anthocyanin pigments from jaboticaba (variety Sabará) and maceration with ethanol acidified with HCl 1.5 mol·L^−1^ (85:15) promoted better extraction, resulting in 25.98 mg·g^−1^ of cyanidin 3-glucoside. Different results could also be explained due to the use of different solvents and extraction methods used. Other factors must also be taken into account, such as the species of fruit, and the geographical location of the plantation.

The effects of the operational conditions (time and temperature) used in the extraction of the compounds and the interaction between them are reported in the Pareto diagram (Figure 1). As shown in Figure 1a, the temperature had a significant effect at the 95% confidence level; which was an effect that influenced the extraction of TP. From the results obtained, it was observed that the amount of recovered TP increased with increasing temperature. The viscosity of the solvent decreased with increasing temperature, which increased the intermolecular interaction, generating a high molecular movement that resulted in greater solubility. This also increased the mass transfer coefficients. Higher temperatures could also cause cell disruption due to increased intracellular pressure, a fact that helps to increase extraction rates [19].

For AA (Figure 1b), it was observed that none of the operational conditions were significant at the 95% confidence level. Therefore, extraction time and temperature were factors that did not significantly influence the final result of AA recovery. For TA (Figure 1c), time was significant in the studied ranges. This variable had a negative influence on extraction, showing that this type of electromagnetic wave could cause anthocyanin degradation, obtaining greater results with the use of shorter extraction times. For C-3-G (Figure 1d), the main anthocyanin of the jaboticaba bark, the same behavior was observed, since the temperature and the interaction between temperature and time were significant and had a negative effect. Despite the observed fact, the temperature provided a good recovery of this compound. Similar behavior was observed by Alara et al. [20] when studying the recovery of bioactive compounds from Phaleria macrocarpa through microwave, using temperatures in the range of 60 °C to 100 °C. These authors reported that the yield recovery increased up to 80 °C, but the amount decreased due to the degradation of the thermolabile constituents and evaporation of the volatile components, above 80 °C.

Akbari et al. [19] reported the importance of finding an equilibrium point between temperature and time, since a longer time could cause a bioactive compound degradation, especially when using water as solvent. According to Chemat and Cravotto [20], water has a higher dielectric constant that could cause a phenomenon called superheating, which occurs when the system absorbs more microwave energy than it can dissipate, and as a consequence the temperature inside the matrix increases. In some cases, this phenomenon can cause the degradation of the compounds of interest. This explains the significant and negative effect of the interaction (extraction time and temperature), as well as the significant and negative effects of temperature and extraction time on the extracted C-3-G content, since once the phenomenon described above is reached, the compound of interest starts to degrade.

The analysis of variance (ANOVA) of the predictive models for bioactive compounds obtained are presented in Table 2. The statistical significance of the models was checked by an F test, and it could be verified that they did not show a lack of fit (F_value_ < F_tabled_).

Predictive models are shown in Equations (1)–(3) to estimate the values of TP, TA, and C-3-G, as a function of statistically significant parameters (*p* < 0.05). The coded variables were temperature (x) and time (y) for the MAE.
TP = 355.11 + 33.75x − 30.55x^2^(1)
TA = 161.91 − 37.88x^2^ − 17.72y + 29.20y^2^(2)
C-3-G = 725.81 − 83.96x − 241.78x^2^ − 111.26xy(3)

Analyzing Figure 2, it was possible to observe that the optimization of the extraction process of the bioactive compounds could be obtained using the extraction time and temperature presented in Table 3. Extraction using the optimized conditions was carried out in order to confirm the efficiency of the models. From the results obtained, it was observed that the values predicted by desirability were in agreement with those found experimentally (Table 3).

The bioactive compounds results were high in a short time, thus, the present research is of great value, as it provides data on the extraction of bioactive compounds from the peel of jaboticaba, whose information is still limited. In addition, the research sought to solve a waste problem that was generated, which were the peels, thus, making use of this waste that has enormous potential as a source of bioactive compounds. Additionally, an innovative form of extraction, MAE, was applied, using water as a solvent, aiming at a lesser impact on the environment.

This result was very positive, because *E. coli* is an important cause of community-acquired infections. Moreover gram-negative bacteria like *E. coli*, could be resistant to antimicrobial action since their cell wall had a protective layer of polysaccharides that hindered the action of antimicrobial agents present in the extracts [21,22]. This microorganism is normally spread by water and foods, however, the major hosts are ruminants, i.e., cattle and sheep [23]. Thereby, the carrageenan-based film with the addition of jaboticaba peels extract could be used as active food packaging material, specifically for meat packaging applications.

In the literature, many authors already reported the antimicrobial activity of jaboticaba fruit, which could be present both in the peel, the pulp, and in the leaves [24,25,26]. This fact was attributed to the high content of phenolic compounds that were known to have antimicrobial properties. Borrás-Linares et al. [27] found that phenolic compounds from plant materials have important antimicrobial property, which normally acts through destabilization of the cell wall. Thus, rupture of cytoplasmic membranes occur. This initiates mechanisms like enzyme inactivation and protein denaturation, which retards bacterial growth and multiplication [28].

Figure 3 shows the FTIR spectra of powdered jaboticaba peels and extract of jaboticaba peels. Through the analysis, it was possible to observe the peak at 1610 cm^−1^, which was more pronounced in powdered jaboticaba peels. This absorption could be attributed to the elongation of the aromatic ring that could be confirmed by the literature [29,30], which reported bands in the region between 1680 and 900 cm^−1^ and normally corresponds to the presence of phenolic compounds that are characterized by an aromatic ring with one or more hydroxy substituents. These bands corroborated what was observed by the total phenolic analysis, and confirmed the presence of bioactive compounds in the powder of the jaboticaba bark, as well as the efficiency of the extraction procedure employed.

Another peak in 1726 and 1728 cm^−1^ for jaboticaba bark extract and jaboticaba peel powder, respectively, could be attributed to the group C=O of the carboxylic acid [30], which was also present in the anthocyanins structure bonds and in the acid gallic that appeared in the peel of jaboticaba. Meanwhile, the spectral region 3398–3437 cm^−1^ could be attributed to the absorbance of hydrogen bonds and the lengthening of the OH bond in relation to carboxylic acids and residual water [31]. Vibrations of 1020 and 1028 cm^−1^ were associated with the elongation of C-O-C bonds in anthocyanins [32].

### 2.2. Biogegradable Films Characterization

As observed, the produced jaboticaba extract had excellent characteristics, so the extract obtained by MAE was produced under optimized conditions and applied to biodegradable films. The proposition of this new packaging brings many benefits and emerges from the concern of a society with environmental issues, and furthermore reflects the characteristics of conscious consumption, aiming at a healthier lifestyle, far from chemical additives and plastics. A current and interesting study revealed that we eventually consume microplastics that come off the packaging that is used to pack meat [33]. The biodegradable films produced CAR-control and CAR-JPE appeared to be homogeneous, uniform, non-brittle, and flexible, and were easily removed from the support (plate) (Figure 4).

The Table 4 shows the values obtained for properties of the carrageenan biodegradable films, with and without the added extract.

The results showed that the film thickness increased with the addition of the jaboticaba peel extract. This was mainly due to the increase in the solid content of the composite films [34]. In addition, it is possible that anthocyanins, present in the extract and with abundant hydroxyl groups, act as bridges and bind strongly to the polymer, forming a network structure through intermolecular interactions, thus, increasing the thickness of the film [29,30].

Hanani et al. [31] studying the incorporation of pomegranate peel powder in fish gelatin films, noticed an increase in the film thickness, due to the insoluble pomegranate particles in the films. Rosa et al. [32] also found an increase in the thickness of carrageenan films with different concentrations of olive leaf extract.

The solubility of the films was an important parameter when the objective was development of the packaging material, as it indicated its water resistance and their biodegradability [35,36]. The carrageenan film water solubility, with and without the added extract, was higher than films with other polymers, as reported in the literature. Gómez-Aldapa et al. [37] reported that the solubility of starch potato films was 28.05%. Francisco et al. [38] developed acetylated cassava starch film and found a solubility at 28.73%. This difference could be explained by the hydrophilicity of carrageenan, which absorbed water and swelled quickly [39]. However, the results were in agreement with those reported by Nazurah et al. [36], for the control film made with carrageenan and for the films with 1% addition of plant oils, which showed solubility in the range of 99.1 to 100%. Ganesan et al. [40] produced carrageenan films with pre-treatment, using different concentrations of KOH and the obtained solubility values that ranged from 77.5 to 92.5%. Using the Tukey test for the parameter solubility, it was possible to verify that it was not significant difference. Nevertheless, the increase in film water-solubility with the addition of the extract could be explained by the interaction between hydroxyl groups, present in the same, with water molecules from the environment further increasing the moisture content [41]. This was because the major compound present in the extract is cyanidin-3-glucoside—a polar compound [42].

It was observed that the films with extract showed a slight increase in the stretching capacity. The inclusion of extract in the carrageenan films induced a reduction in its tensile strength, however, the change in this parameter was not significant. As reported by Chi et al. [13], the decrease in tensile strength might be attributed to the formation of defect points and stress concentration points in the films. Rosa et al. [34] also reported a reduction in the tensile strength of carrageenan films, with different concentrations of the incorporated olive leaf extract.

The effect of addition of JPE into the color of biodegradable films is also shown in Table 4. From this data, it was clear that the addition of extract resulted in higher reddish and yellowness (increase of *a** and *b**) of the films, increasing the color difference between films, with and without extract (control). The difference between the parameters *L**, *a**, and *b** was statistically significant, and the films differed by 96.50%. The JPE improved the light barrier of the films, giving the carrageenan extra protection against oxidative processes. Gurak et al. [2] also had its jaboticaba extracts classified as reddish. Recently, several researchers reported that the addition of antioxidant compounds could influence the optical properties of film, by decreasing the lightness and film transparency values, and increasing the reddish and yellowness values, correlated with the increasing content of the antimicrobial agent [43].

## 3. Materials and Methods

### 3.1. Reagents

Carrageenan, glycerol, 2,2-diphenyl-1-picrylhydrazyl (DPPH), Folin Ciocalteu’s phenol, anhydrous sodium carbonate, and gallic acid were of analytical grade. Water, acetonitrile, formic acid, and cyanidin-3-glucoside were of HPLC grade. For the antimicrobial analysis—Nutrient, Müller-Hinton broth, and Kanamycin were used. All reagents were purchased from Sigma Aldrich (St. Louis, MO, USA). The bacteria strain used in the antimicrobial experiment was *Escherichia coli* K12 (ATCC 10798) obtained from Cederlane (Burlington, ON, Canada).

### 3.2. Sample Preparation

The jaboticaba fruits (*Myrciaria jaboticaba*) were collected from a farm located at São José do Cedro, Santa Catarina, Brazil. The fruits were washed in running water and shelled manually. The peels were sanitized with a commercial solution of 2% sodium hypochlorite and rinsed in sterilized distilled water, then stored at −18 °C. The peels were dried in a freeze-dryer at −50 °C for 48 h, and stored in vacuum sealed bags. The jaboticaba peels were ground to a fine powder by an analytical mill (IKA, A11, Darmstadt, Germany), and the fraction retained in the 60 mesh sieve was used in order to standardize the particles.

### 3.3. Extraction Procedures

Approximately, 0.5 g jaboticaba peels powder were added in a quartz tube and homogenized in 25 mL of solvent (distilled water, liquid phase) and the microwave-assisted extraction (MAE) was performed using a closed multi-mode Mini-WAVE microwave unit (SCP Science, Baie-D’Urfé, QC, Canada). The microwave unit was equipped with power, temperature, and time controllers, which allowed the application of a frequency and power of irradiation of 2.45 GHz and 1000 W, respectively. After the MAE, the samples were filtered and the extracts were characterized.

The MAE was based on a central composite rotational design (CCRD) to determine the effect of variables (Table 5) on the extracts properties (total phenolic compounds, antioxidant potential, total anthocyanins, and cyanidin-3-glucoside—all data reported was the average of the triplicate analyses). Generally, high temperatures are related to more efficient extractions, but there is a dichotomy that high temperatures simultaneously increase the rate of degradation of the compounds of interest. In this context, the time spent on extraction was decisive. Therefore, high temperature and short term extraction conditions were used successfully to delay degradation and increase the extraction efficiency of the compounds of interest [44]. The ranges used for the variables were determined from preliminary tests and literature, which reported that the microwave extraction technique stood out for it is fast transfer of energy, accelerating the process of mass transfer from the plant matrix to the solvent. CCRD also was used as a tool for optimization of conditions to increase the presence of bioactive compounds, based on modeling and the desirability function.

### 3.4. Bioactive Compounds

The total phenolic (TP) quantified by the spectrophotometric method adapted from the work of Singleton and Rossi [45] was used to quantify the total phenolic content with Folin-Ciocalteu reagent and standard gallic acid. Total phenolics were quantified using a standard curve with different concentrations (72–1800 mg L^−1^) of gallic acid.

Antioxidant activity (AA) was determined with the method developed by Brand-Williams et al. [46]. DPPH (6.10^−5^ M) was prepared with the DPPH standard and methanol. The absorbance of blank as well as solutions with extracts was measured using a spectrophotometer (Ultraspec1000, Amersham Pharmacia Biotech, USA) at 517 nm. *Total Anthocyanins* (TA) content was estimated by the method proposed by Fuleki and Francis [47]. Absorbance of the extract solution was measured at a wavelength of 520 nm using a spectrophotometer (Ultraspec1000, Amersham Pharmacia Biotech, Chiltern, Bucks, England).

The cyanidin 3-glucoside was the majority anthocyanin pigment of the skins, therefore, the quantitative analyses of C-3-G in JPE were performed by high performance liquid chromatography—HPLC, using an Agilent 1100 (Santa Clara, CA, USA), equipped with a variable wavelength detector (VWD). The isocratic mobile phase consisted of water/acetonitrile/formic acid solvents (80/10/10 *v/v/v*). The extracts obtained from the assisted microwave were filtered through a 0.45 mm syringe filter and injected directly into the HPLC. The concentrations of C-3-G in the extracts were quantified using a standard curve, with concentrations ranging from 0.05 to 0.4 mg L^−1^.

### 3.5. Antimicrobial Inhibition (AI)

The antimicrobial inhibition potential of the extract obtained by the optimal condition, was tested against a gram-negative bacteria *Escherichia coli* K12 (ATCC 10798), following the micro-dilution method of Clinical and Laboratory Standards Institute (CSLI, 2015). The lyophilized extract was obtained using a freeze dryer (Gamma 1-16 LSC, Christ, Osterode am Harz, Germany) for 48 h. The sample was tested at concentrations of 50 mg mL^−1^. *E. coli* was cultured in nutrient broth at 37 °C for 24 h, in shaken flasks. To 96-well microliter plates, 135 μL of extract, 145 μL of sterile Muller-Hinton broth, and 20 μL of the *E. coli* culture were added. A control (inoculum without extract) was included on a microplate. The microplate was incubated at 37 °C for 16 h in a microplate spectrophotometer, with controlled temperature (PowerWave XS, Biotek, Winooski, VT, USA). The contents of the wells were mixed before reading the absorbances at 620 nm (OD620). Percent growth inhibition was calculated by Equation (4).
(4)$$AI=\left[1-\left(\frac{OD_{JPE.2}-OD_{JPE.1}}{OD_{control.2}-OD_{control.1}}\right)\right]\times 100$$
where *AI* is the inhibition (%), *OD_JPE._*_2_ is the OD620 for the sample after the incubation period, *OD_JPE._*_1_ is the OD620 for the sample before the incubation period, *OD_control._*_2_ is the OD620 for control after the incubation period, and *OD_control._*_1_ is the OD620 for the control before the incubation period. Each experiment was repeated three times.

### 3.6. Fourier Transform Infrared Spectroscopy (FTIR-ATR)

The FTIR spectra of the jaboticaba peel powder and jaboticaba peel extract (JPE) were recorded using an infrared spectrometer with ATR (Perkin-Elmer, UATR Two, Waltham, MA, USA), in the range of 400 to 4000 cm^−1^. It was used with 32 scans per spectrum with a resolution of 4 cm^−1^.

### 3.7. Biodegradable Film Formation

Carrageenan films were developed by the casting method. To prepare the filmogenic solutions, 0.5 g of biopolymer powder was dissolved in 30 mL of distilled water and heated at 70 °C for 15 min, using a hot plate and a magnetic stirrer (Cimarec2, Thermolyne, Luoyang, China). During agitation, glycerol was added into carrageenan solution, at a concentration of 60% *w/w* based on the weight of the polymer, the similarity of the experimental procedure employed by Rosa et al. [34]. After this, the lyophilized extract of jaboticaba peels was added to the already cooled solution at a concentration of 20 g L^−1^. Finally, the film-forming solution was poured into polystyrene petri dishes (90 mm diameter) and biodegradable films were obtained through solvent evaporation using a convective dryer. After drying, the films (CAR-JPE) were manually peeled off from the plates and conditioned in a relative humidity of 50% at room temperature for 48 h, for further analysis. A biodegradable film of carrageenan without the addition of jaboticaba peels extract was produced as control, called CAR-control.

### 3.8. Biodegradable Film Characterization

#### 3.8.1. Film Thickness

The thickness was measured at ten different locations on the film, using a digital micrometer (Insize-IP65, Asia). The accuracy of the micrometer was 0.001 mm.

#### 3.8.2. Color Measurement

The color of the film samples was determined using a spectrophotometer (Konica Minolta, CM-2600D, Tokyo, Japan). The parameters *L* (lightness/brightness), *a* (redness/greenness), and *b* (yellowness/blueness) were measured and the total color difference (Δ*E*) was calculated as follows:(5)$$\Delta E=\sqrt{\left(L^{*}-L_{s}^{*}\right)2+\left(a^{*}-a_{s}^{*}\right)2+\left(b^{*}-b_{s}^{*}\right)2}$$
where the $L^{*}$, $a^{*}$, and $b^{*}$ parameters are the values of the control (CAR-control) and the parameters with the sub-index *s* refers to the CAR-JPE sample.

#### 3.8.3. Solubility in Water

The water solubility of the film samples was determined according to the method described by Gontard and Guilbert [48]. Film samples were dried at 105 °C for 24 h and weighed to determine their initial dry weight. Then, the samples were uniformly cut in 2 cm diameter circles and placed in recipients with 50 mL of distilled water. The system was kept under agitation (100 rpm), at 20 °C for 24 h, using an incubator shaker (Solab, SL 223, Piracicaba, SP, Brazil). The undissolved films were filtered and dried at 105 °C for 24 h, to determine their final dry weight. The water solubility (WS%) was calculated using the following equation:(6)$$WS\left(\%\right)=\left[\frac{W_{0}-W_{f}}{W_{0}}\right]\times 100$$
where $W_{0}$ is the initial dry weight of the film and $W_{f}$ is the final weight of the dried undissolved film.

#### 3.8.4. Mechanical Properties

Tensile strength (TS) and elongation percentage (E) at break point were measured uniaxially, by stretching the specimen in one direction, using a Universal Testing Machine (Instron 4502, Norwood, MA, USA), according to the ASTM Standard D882-09 [49]. Samples were clamped and deformed under tensile loading, using a 50 N load cell with an initial grip separation of 25 mm and a cross-head speed of 50 mm·min^−1^.

### 3.9. Analysis of Results

The results of the biodegradable film properties were analyzed with the statistical software Statistica^®^, version 10.0 (SAS Institute, Cary, NC, USA). The mean comparisons were carried out by the Tukey test, to determine possible significant differences among the treatments at a 95% confidence level, and analysis of variance (ANOVA).

## 4. Conclusions

Jaboticaba peels are a residue with enormous potential to add value, and in this study, different extraction conditions were used by microwave-assisted extraction (MAE). From the desirability tool, it was demonstrated that the increase in temperature and the shorter time was the best choice to obtain a higher bioactive action, in which the optimum conditions were 80 °C and 1 min. The results indicated that the values predicted by the models were in agreement with those found experimentally. The jaboticaba peel extract (JPE) showed a combined antimicrobial and antioxidant actions. JPE was highly effective against *E. coli* and showed excellent values of phenolic compounds, anthocyanins, cyanidin 3-glucoside, and antioxidant activity, indicating its potential as a source of bioactive compounds and for use as a functional ingredient in food packaging and to provide an added value to by-products.

The biodegradable films based on carrageenan containing jaboticaba peel extract developed in this study exhibited good properties for food packaging. The impact of the addition of JPE on the mechanical properties of the biodegradable carrageenan films was positive, as it increased the elongation at break and decreased the tensile strength. For the color, analysis differentiated from the control in 96.50%. In addition, the antioxidant and antimicrobial properties of the extract suggest that this biodegradable film has a strong potential for use as active packaging systems. Thus, this new material could help the food industry to potentially safely extend the shelf life of food products, as well as reduce the consumption of traditional plastic films.

## Figures and Tables

**Figure 1 molecules-25-05563-f001:**
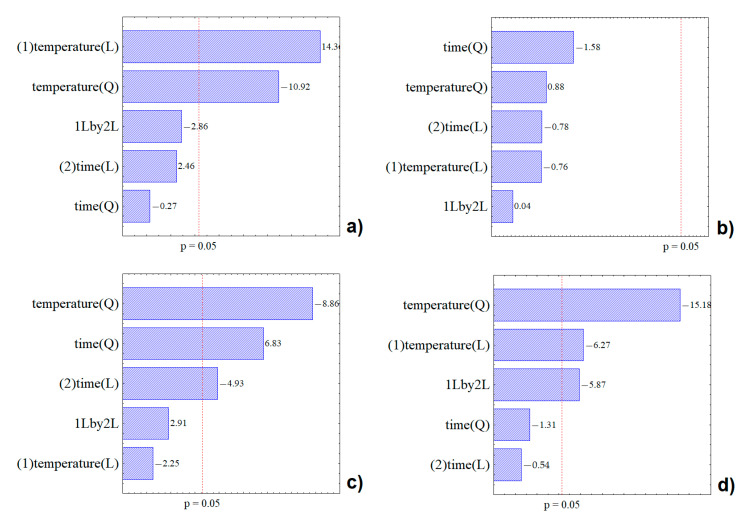
Pareto diagram of the estimated effects on TP (**a**), AA (**b**), TA (**c**), and C-3-G (**d**).

**Figure 2 molecules-25-05563-f002:**
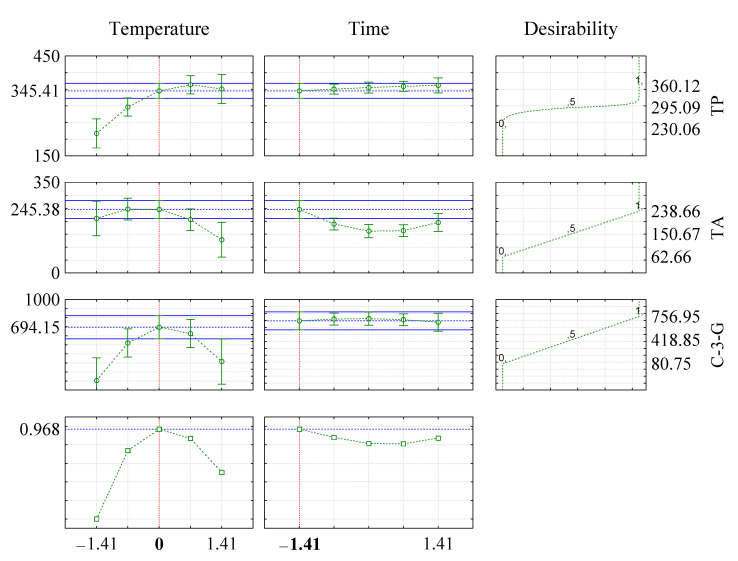
Desirability function for the optimizing conditions for the extraction of bioactive compounds from jaboticaba peels.

**Figure 3 molecules-25-05563-f003:**
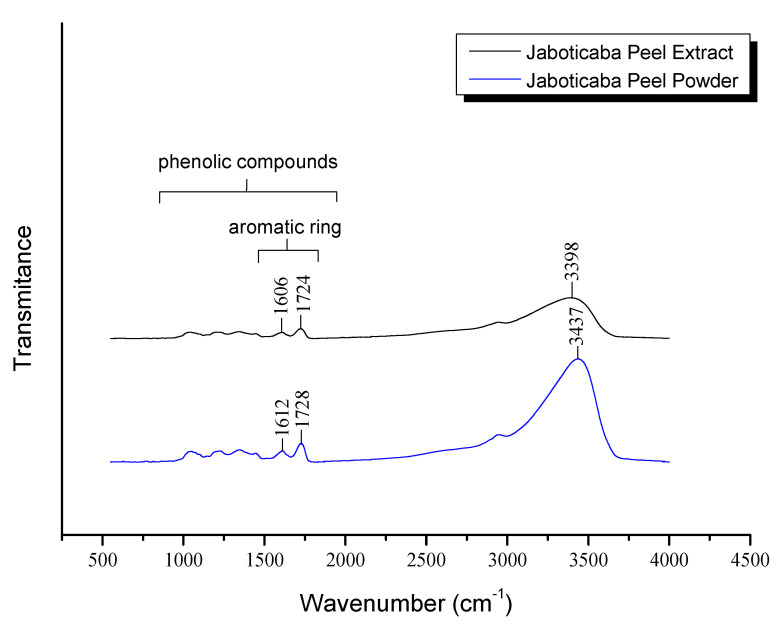
FTIR spectra of the jaboticaba peel powder and extract.

**Figure 4 molecules-25-05563-f004:**
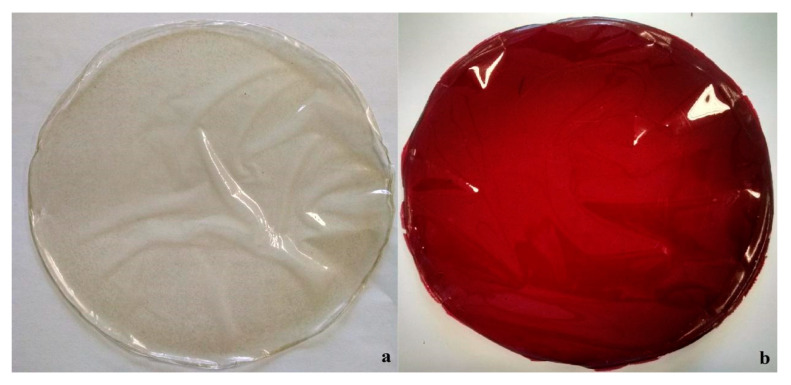
Biodegradable carrageenan films appearance: (**a**) CAR-control and (**b**) CAR-JPE.

**Table 1 molecules-25-05563-t001:** Total phenolic content (TP), antioxidant activity (AA), total anthocyanin (TA), and cyanidin 3-glucoside (C-3-G) of the extracts obtained from jaboticaba peels.

	x (°C)	y (min)	TP (mg_GAE_·g^−1^) (d.b.)	AA (%)	TA (mg·100g^−1^) (d.b.)	C-3-G (mg_cyan_·g^−1^) (d.b.)
1	−1(60)	–1(2)	283.52 ± 12.22	94.5 ± 0.05	238.66 ± 5.86	435.91 ± 1.16
2	−1(60)	1(6)	319.82 ± 2.43	93.9 ± 0	141.79 ± 4.16	656.97 ± 0.93
3	1(100)	–1(2)	347.84 ± 3.76	92.8 ± 0.9	164.11 ± 2.23	526.41 ± 0.44
4	1(100)	1(6)	346.08 ± 4.23	92.4 ± 0.8	126.41 ± 4.53	302.41 ± 0.35
5	−1.41(52)	0(4)	230.06 ± 0.44	94.9 ± 0.02	62.66 ± 1.41	369.01 ± 0.16
6	1.41(108)	0(4)	356.93 ± 2.48	94.8 ± 0.1	80.55 ± 2.16	80.75 ± 0.03
7	0(80)	−1.41(1)	348.92 ± 1.75	92.6 ± 0.6	208.33 ± 7.04	686.05 ± 0.22
8	0(80)	1.41(7)	357.26 ± 0.60	90.9 ± 0.6	203.25 ± 1.66	646.89 ± 0.60
9	0(80)	0(4)	347.58 ± 2.57	92.4 ± 0.2	154.54 ± 2.63	683.69 ± 12.74
10	0(80)	0(4)	357.65 ± 2.12	95.4 ± 0.2	157.69 ± 2.91	554.30 ± 5.48
11	0(80)	0(4)	360.12 ± 6.71	93.6 ± 0.4	173.49 ± 4.68	736.82 ± 5.56

Data reported are average of 3 replicates and ± mean deviation.

**Table 2 molecules-25-05563-t002:** ANOVA for TP, TA, and C-3-G from jaboticaba peels.

	Sum of Squares	Degrees of Freedom	F_value_	F_tabled_	R^2^	R^2^ Adjusted
TP						
Regression	15,433.02	1	13.51	161.44	0.93	0.99
Residual	1141.63	1				
Lack of fit	1053.34	3	7.95	19.16		
Pure error	88.29	2				
Total	16,574.65	10				
TA						
Regression	22,078.16	3	3.95	9.27	0.79	0.99
Residual	5581.62	3				
Lack of fit	5375.34	3	17.37	19.16		
Pure error	206.28	2				
Total	27,659.78	10				
C-3-G						
Regression	451,997.1	1	55.11	161.44	0.98	0.99
Residual	8201.3	1				
Lack of fit	5336.1	3	1.24	19.16		
Pure error	2865.2	2				
Total	460,198.4	10				

**Table 3 molecules-25-05563-t003:** Optimized conditions for the extraction of TP, TA, and C-3-G using MAE, corresponding to the predicted and observed yields.

Exp Conditions	TP, Pred Response (mg_GAE_·g^−1^) (d.b.)	TP (mg_GAE_·g^−1^) (d.b.)	TA, Pred Response (mg·100g^−1^) (d.b.)	TA (mg·100g^−1^) (d.b.)	C-3-G, Pred Response (mg_cyan_·g^−1^) (d.b.)	C-3-G (mg_cyan_·g^−1^) (d.b.)
80 °C, 1 min	345.41	383.81 ± 7.41	245.38	207.96 ± 7.82	694.15	789.25 ± 3.02

**Table 4 molecules-25-05563-t004:** Thickness, solubility, mechanical properties, and optical properties of films.

Film	Thickness (mm)	Solubility (%)	Elongation at Break (%)	Tensile Strength (MPa)
CAR–control	0.032 ^a^ ± 0.0048	79.02 ^a^ ± 4.76	21.30 ^a^ ± 3.58	10.69 ^a^ ± 1.61
CAR–JPE	0.054 ^b^ ± 0.0055	82.84 ^a^ ± 10.57	28.26 ^a^ ± 3.69	6.08 ^a^ ± 0.33
**Optical Properties**				
Film	*L**	*a**	*b**	Δ*E*
CAR–control	94.49 ^a^ ± 0.21	−0.145 ^a^ ± 0.03	3.32 ^a^ ± 0.30	-
CAR–JPE	19.96 ^b^ ± 1.08	52.77 ^b^ ± 1.10	34.26 ^b^ ± 1.86	96.50

Data reported are the average values and ± mean deviation. Note: Different letters in the exponent (a and b) represent significant differences (*p* < 0.05) between the mean obtained by the Tukey test. CAR-control = carrageenan biodegradable film; and CAR-JPE = carrageenan biodegradable film with jaboticaba peel extract. *L** = lightness/brightness; *a** = redness/greenness; *b** = yellowness/blueness; and Δ*E* = total color difference.

**Table 5 molecules-25-05563-t005:** Independent variables of the experimental design and the extraction conditions.

**Variables**	**Levels**				
	−1.41	−1	0	+1	+1.41
Temperature (°C)	51.8	60	80	100	108.2
Time (min)	1.18	2	4	6	6.82

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
