# Peer review of "Carrageenan-Based Films Incorporated with Jaboticaba Peel Extract: An Innovative Material for Active Food Packaging"

_molecules, 2020, doi:10.3390/molecules25235563_

Round 1
Reviewer 1 Report
The manuscript is well written and covers both extraction and characterization of bioactive compounds from jaboticaba peels, as well as production and evaluation of environmentally friendly carrageenan films with potential use in active packaging. Optimal conditions for microwave assisted extraction have been selected and used, although the explanation for selection of these conditions covered almost half of Results and Discussions, which is not appropriate. At least Anova Table (3) could be moved to supplementary material (as I’ve found in many other published papers) and more detailed discussions on properties findings should be inserted in the manuscript as suggested below.
Abstract
- As only one composition has been used for obtaining carrageenan/JPE film, singular (film, not films) should be used in “The carrageenan films incorporated with JPE were manageable, homogeneous and the presence of JPE into films …”;
– Please remove significantly from “increased significantly the thickness” as results do not support a significant increase.
Keywords:
As no biodegradability tests have been performed for the obtained film described in this manuscript, “biodegradable packaging” cannot be used as keyword. I suggest using “microwave assisted extraction”, carrageenan film, jaboticaba peel”
Page 1, lines 35-36
“This fruit is widely consumed in Brazil and is popularly known as Brazilian Grape Tree” – not the fruit, but the tree producing jaboticaba fruits, is named Brazilian Grape Tree?
Page 2, lines 46-48
“The extraction condition applied for its recovery is considered a critical factor to achieve the optimum quality and quantity of the bioactive compounds, such as the different process parameters (type of solvent, extraction method, temperature and time) can significantly influence its quality.” – Please reformulate this sentence, is difficult to understand it.
Page 2, lines 78-79
“Furthermore, to develop carrageenan-based biodegradable films with the incorporation of jaboticaba peel extract (JPE).” – Please reformulate this sentence, is difficult to understand it.
Page 4, lines 137-139
What do the authors mean with “The significant effect of the interaction also shows that in shorter times the effect of temperature on the extracted C-3-G content was different when the time was greater.”?
Page 7, lines 186-187
Peel, pulp and leaves of other fruits/vegetables? what exactly do you refer when mentioning “In the literature many authors have already reported the antimicrobial activity that can be presented both in the peel, pulp and in the leaves [26–28]. “
Page 7, lines 187-189
There is a repetition in “This fact is attributed to the high content of phenolic compounds that are known to have antimicrobial properties. This fact is attributed to the high content of phenolic compounds that are known to have antimicrobial properties.”
Page 8, line 209 - (De Souza et al., 2015) should be properly reported as other references.
Page 9, lines 236 - 237
This phrase should be completed “This explained because the interstitial spacing between polymeric chains in the film matrix was increased by the dispersed extracts [8].”
Page 9, lines 252-253
What is the meaning of “Through the solubility Tukey test, it is possible to verify that it was not significant.”?
Page 9, lines 248-250
Please mention the type of control film in “However, the results are in agreement with those reported by Nazurah R. & Nur Hanani [38] for the control film and for the films with 1% addition of plant oils which showed solubility in the range of 99.1 to 100%. “
Page 9, lines 258-259
An explanation of mechanical properties modification should be mentioned, not just self citation of the authors’ previous work.
The phrase should be completed: “The inclusion of extract in carrageenan films induced a reduction in its tensile strength, however, the change in this parameter was not.”
Page 10, lines 268-271
Please explain role of ANTIMICROBIAL agent in modification of optical properties. Reference 45 suggested the influence of ANTIOXIDANT activity over these changes.
Conclusions – should address more specific to the properties results presented in the manuscript as in the current form are too general.
As no experiments have been performed by authors regarding shelf life of food products, authors should mention “potentially contribution to food shelf-life extension”
Author Response
Please find attached the answers.

Reviewer 2 Report
The paper deals with the investigation of the jaboticaba peel extract (JPE) bioactive potential and proposes carrageenan films with JPE incorporated as innovative food packaging materials. The study is focused on the optimization of process parameters in the extraction procedure - microwave assisted extraction (MAE) and on the characterization of bioactive jaboticaba peel extract (JPE) by assessing total phenolic content (TP), antioxidant activity (AA), total anthocyanin (TA) and cyaniding 3-glucoside (C-3-G) content and antimicrobial activity against E. coli. For the carrageenan film incorporated with JPE - the thickness, solubility, mechanical and optical properties have been studied.
I consider that the work reported by the authors is original, containing interesting experimental data and pertinent discussions, being properly structured throughout the paper. However, I appreciate that the paper needs some improvements.
The authors should emphasize more on the study of the carrageenan films with JPE incorporated and their characterization, novelty, advantages – as the title suggests, since there is already available literature on the extraction and properties of bioactive compounds from jaboticaba peel extracts.
Generally, the paper seems enough-documented in citing similar results obtained, but the authors should support better and discuss more detailed their own results and justify their statements for strengthening their work.
English editing should be checked.
- The paragraph at page 2, lines 60-61 „to maximize their additional characteristics, such as their antimicrobial capacities.” should be rephrased, adding the antioxidant properties, since polyphenols and anthocyanins are compounds known especially for their antioxidant activity.
- Table 2 at page 3 should be Table 1.
- The „operating conditions” reported in the Pareto diagrams represented in Figure 2 should be further explained in order to clarify the variables analyzed and their interactions, mentioning the exact significance of the parameters on the graph.
- At page 7, lines 177-179 – the evaluation of antimicrobial activity is only mentioned in the text as having an efficiency of 100%, but there are no experimental results to support this statement.
- The statement at page 7, lines 187-189: „This fact is attributed to the high content of phenolic compounds that are known to have antimicrobial properties.” is repeated and should be rephrased and further argumented.
- A further detailed discussion of the FTIR spectra is needed, with more information on the jaboticaba peel powder/extract and carrageenan regarding chemical composition and evidence of functional groups. A comparative study with the spectra corresponding to the obtained films - CAR-control and CAR-JPE would be more relevant.
- At page 8, line 207 – the term „signal” should be avoided.
- Table 6 at page 11 was not mentioned in text.
- At page 11, line 335: „3.5. Antimicrobial Inhibition (AI)” – should be subtitle.
Author Response
Please find attached the answers.

Reviewer 3 Report
The manuscript entitled "Carrageenan-based films incorporated with jaboticaba peel extract: an innovative material for active food packaging", needs to be revised before it can be accepted for publication. Some recommendations are as the following:
In the introduction, the novelty of the current study should be emphasized more clearly.
In this study, only the DPPH method is used to evaluate antioxidant activity. The measurement of antioxidant activity, in the case of multifunctional or complex multiphase systems, cannot be evaluated satisfactorily by a single method. Even the methods based on the same principle can show several important differences in their response to antioxidants. Therefore, I highly recommend applying several different assays to evaluate antioxidant activities to obtain the full picture.
How the authors identified the anthocyanins if they just used HPLC? Mass spectrometry analysis is required to confirm the Cya-3-Glu.
Change all XoC for X oC
Figures: All the figures must be improved in terms of resolution. Figure 1 needs great improvement. Figure 2, the resolution must be improved, and also the variables presented in the axes must be replaced and standardized. Figure 3 needs an improvement considering the resolution. Figure 4 also need improvement on the resolution as well as the font must be standardized regarding the size and type, the unit must be corrected.
Author Response
Please find attached the answers.

Round 2
Reviewer 2 Report
I consider that the comments were answered satisfactorily, with the following remarks:
- The authors added equation 4 in section 3.5. to justify the result of 100% antimicrobial efficiency – as requested, but why the result of 100% inhibition of E. coli was deleted from text (page 7-line 178)?
- The wavenumbers for the FTIR spectra absorption bands have changed in Figure 3 - manuscript-v2 and they don’t correspond anymore with the ones discussed in text (1610, 1726 cm-1); authors should return to the old figure or change the values in the text. Also, terms such as “elongation” and “lengthening” are not adequate for FTIR spectra discussion, stretching was just fine.